# Inhibitory Activity of Hydroxypropyl Methylcellulose on Rhinovirus and Influenza A Virus Infection of Human Nasal Epithelial Cells

**DOI:** 10.3390/v17030376

**Published:** 2025-03-06

**Authors:** Hsiao-Hui Ong, YongChiat Wong, Jayant Khanolkar, Belinda Paine, Daniel Wood, Jing Liu, Mark Thong, Vincent T. Chow, De-Yun Wang

**Affiliations:** 1Department of Otolaryngology, Yong Loo Lin School of Medicine, National University of Singapore, Singapore 117545, Singapore; entohh@nus.edu.sg (H.-H.O.); entliuj@nus.edu.sg (J.L.); 2Infectious Diseases Translational Research Programme, Yong Loo Lin School of Medicine, National University of Singapore, Singapore 117545, Singapore; micctk@nus.edu.sg; 3Personal Health Care, Procter & Gamble International Operations SA Singapore Branch, Singapore 138547, Singapore; wong.y.3@pg.com; 4Personal Health Care, Procter & Gamble UK, Reading RG2 0RX, UK; jkhanolkar@gmail.com (J.K.); paine.b.1@pg.com (B.P.); daniel.guy.wood@gmail.com (D.W.); 5Department of Otolaryngology-Head & Neck Surgery, National University Health System, Singapore 119228, Singapore; mark_thong@nuhs.edu.sg; 6Department of Microbiology and Immunology, Yong Loo Lin School of Medicine, National University of Singapore, Singapore 117545, Singapore

**Keywords:** nasal epithelium, rhinovirus, influenza A viruses, hydroxypropyl methylcellulose, HPMC, acidic buffer

## Abstract

The nasal epithelium is the primary site for entry of respiratory viruses. In comparison to oral administration, nasal drug applications directed locally to the site of infection can serve as early interventional barriers against respiratory virus pathogenesis by limiting viral spread in the upper airway. Experiments on the diffusion of methylene blue and nanoparticles in both water and low pH conditions revealed that hydroxypropyl methylcellulose (HPMC) can act as an effective physical barrier. This study also evaluated the activity of HPMC as a barrier against common respiratory viruses, i.e., rhinovirus (RV) and influenza A virus (IAV) using the in vitro human nasal epithelial cell (hNEC) model. Utilizing the hNEC infection model, we assessed the protective effects of HPMC in pH 3.5 and pH 7 buffers against RV and IAV. Acidic and pH-neutral buffers and HPMC dissolved in acidic and pH-neutral buffers were administered for 4 h prior to virus infection and at 4 h post-infection (hpi). The apical supernatant was harvested at 24 hpi to determine the viral loads of RV and IAV (H1N1 and H3N2). HPMC was demonstrated to exert protective effects in the infected hNECs independent of acidic pH. Pre-treatment with HPMC in acidic buffer significantly diminished viral loads for both RV and IAV infections of hNECs. Similarly, direct treatment of HPMC in acidic buffer after infection (4 hpi) also effectively decreased viral loads of both RV and IAV. Moreover, treatment using HPMC in acidic buffer before or after infection did not affect the epithelial integrity and ciliary function of hNECs. This study demonstrates the protective effects of HPMC in acidic buffer against RV and IAV infections of the human nasal epithelium.

## 1. Introduction

The nasal epithelium in the upper respiratory tract is the first point-of-contact for the entry of respiratory viruses. Upon infection, the airway epithelium of the nasal mucosa serves as a physical barrier by secretion of mucus which traps pathogens. With the beating ciliary movement, mucus is discharged into the nasopharynx from where it is eventually swallowed [1,2]. Muco-ciliary clearance and immune responses elicited by the nasal epithelium protect the host from pathogens. Therefore, treatment or early intervention to enhance the nasal barrier function to counter respiratory virus entry is an appealing approach. It is critical to develop rapid, simple, non-specific, and safe prophylaxis and interventions to augment the nasal barrier function against respiratory virus infection.

Nasal administration at the local site of infection can achieve a quick and direct effect on virus pathogenesis by confining virus spread in the upper airway. Several nasal sprays are commercially available as prophylactic agents, for early intervention against respiratory infections such as the common cold and influenza, and for drug delivery to the nasal cavity [3,4,5]. These commercial nasal sprays aim to block virus entry and transmission within the nasal cavity, the primary site of infection. When administered prior to the peak of viral replication, early intervention against upper respiratory infections (URTIs) can reduce the progression of the infection and thereby reduce viral transmission and disease severity [6].

Hydroxypropyl methyl cellulose (HPMC; CAS No. 9004-65-3) is an odorless, tasteless, white to slightly off-white, fibrous or granular, free-flowing powder that is a derivative of cellulose, the most abundant polymer in nature. It has a wide range of clinical and other applications ranging from drug delivery (to the eye, airway, colon, and skin) to its use as a food ingredient [7,8,9,10]. When insufflated nasally, HPMC swells and forms a physical gel barrier against offending inhaled noxious or allergenic agents, thus reducing the progression of nasal symptoms. A study on HPMC gel matrix reported that HPMC pre-treatment inhibited SARS-CoV-2 infection and that HPMC treatment after SARS-CoV-2 infection inhibited virus release in Vero monkey kidney cells, likely by providing a physical barrier against infection [11]. However, there are limited studies on the effects of HPMC matrix gel on common respiratory viruses such as the common cold virus (rhinovirus) and influenza virus using a clinically relevant human nasal model.

Human rhinoviruses (RV) are the most common cause of URTIs, followed by the more pathogenic influenza A virus (IAV) during and even before the COVID-19 pandemic [12,13,14,15]. The effect of acidity on respiratory viruses, including RV and IAV, has long been a topic of research [16,17]. It was demonstrated that acid has a physical action against respiratory viruses that destabilizes the virus structure, rendering the virus non-infectious [18]. Accordingly, a lower pH of 3.5 to 4.5 was reported to reduce RV and IAV infection in vitro (cell lines) and in vivo (humans, ferrets) [19,20].

As such, nasopharyngeal acidity may have profound implications for virus transmission and intervention strategies. Inactivation of viruses using acidic ozone water and acidic solutions in the form of dry fog have been explored [21,22]. These studies have led to potential intervention strategies on the overlooked role of expiratory aerosol acidity against airborne virus inactivation which may be crucial in minimizing virus transmission especially during pandemic periods, e.g., COVID-19 and influenza [23,24]. However, the impact of acidity on respiratory viruses in vivo may be limited to a transient decrease in pH. The normal nasal pH is 6.3 and can undergo a slight and transient decrease due to the introduction of acidic solutions, but the baseline pH is physiologically restored by nasal homeostatic mechanisms within 15 min after administration [25].

There are limited studies on physiologically relevant human models to investigate the effect of physical barriers against RV and IAV infections. Therefore, the objective of our study was to investigate the effect of HPMC, which serves as a physical barrier on the exterior of the nasal epithelium against RV and IAV by harnessing our previously established in vitro infection model of human nasal epithelial cells or hNECs [26,27,28,29].

## 2. Materials and Methods

### 2.1. Derivation of Human Nasal Epithelial Stem/Progenitor Cells and In Vitro Differentiation of hNECs

Approval to conduct this study was obtained from the National Healthcare Group Domain-Specific Board of Singapore (DSRB code D/11/228) and the Institutional Review Board of the National University of Singapore (IRB code 13-509). The human nasal epithelial stem/progenitor cells (hNESPCs) were derived from tissue biopsies of twelve subjects who underwent septal plastic surgery at the National University Hospital, Singapore. The medical backgrounds of the donors are summarized in Appendix A. The subjects were not on oral or topical corticosteroid medications one month prior to surgery and had no viral infection at the time of surgery. Detailed procedures for the hNEC cultures are provided in the Appendix A.

### 2.2. Virus Infection of Fully Differentiated hNECs

The HRV16 strain 11757 (ATCC VR-283) and IAV H3N2 (A/Aichi/2/1968) strain were obtained from the American Type Culture Collection (ATCC, Manassas, VA, USA). The IAV H1N1 (A/Singapore/G2-25.1/2014) strain is a clinical isolate from the Department of Microbiology and Immunology, National University of Singapore [30]. The hNECs were washed once using 1× dPBS for 10 min at 37 °C. HRV16 (MOI of 2.5), H1N1 (MOI of 0.01), and H3N2 (MOI of 0.01) were used for infection of hNECs. Detailed procedures can be found in the Appendix A.

### 2.3. Viral Titer Quantification Using Plaque Assay

Viral titer quantification using plaque assay was performed as described previously [26,27]. HeLa cells or MDCK cells were used to quantify RV or IAV titers, respectively. Detailed procedures can be found in the Appendix A.

### 2.4. HPMC Preparation and Treatment of hNECs

HPMC (1 g) was dissolved in 100 mL of 1× dPBS (pH 3.5, pH 5 or pH 7) to obtain a final weight of 100 g (1% HPMC). The 1% HPMC was further diluted in 1× dPBS (at pH 3.5, pH 5 or pH 7) at 2× dilution to obtain 0.5% HPMC. To assess cell viability and tolerability of HPMC, 100 µL of 0.5% HPMC was added to the apical compartment of hNECs for 1 day and 1 week. For pre-treatment, 100 µL of 0.5% HPMC was added to hNECs at 4 h prior to infection. Virus inoculum was slowly added onto HPMC solution or acidic buffer alone at 0 h post-infection (hpi). One h after virus inoculation, the solution was removed by gentle aspiration from the apical side of hNECs. The hNECs were then harvested at 24 hpi. For post-infection treatment, 100 µL of 0.5% HPMC was added directly onto hNECs at 4 hpi for 4 h. After treatment for 4 h, the solution was removed by gentle aspiration from the apical side of hNECs. The hNECs were then harvested at 24 hpi.

### 2.5. Liquid Dye Barrier Assessment

To evaluate the physical barrier effect, 1 mL each of the four test formulations, i.e., water only (control), pH 3.5 buffer only (buffered to pH 3.5 with L-pyroglutamic acid, succinic acid and disodium succinate), 1% HPMC in water, or 1% HPMC in pH 3.5 buffer (Vicks First Defence Nasal Spray, Procter & Gamble Manufacturing, Gross-Gerau, Germany) were applied to a 100-µm nylon filter scaffold membrane in a unjacketed, free-standing Franz type diffusion cell (10 mL volume with 2 cm^2^ surface area; PermeGear, Hellertown, PA, USA). Deionized water was used as the receiver fluid. Methylene blue dye (0.5 mL, 0.02% *w*/*w*; Merck, Darmstadt, Germany) was then slowly applied above the test formulation. The potential diffusion of the dye was then video-recorded over a period of 12 min to mimic the rate of muco-ciliary clearance.

### 2.6. Solid Particle Barrier Assessment

Each of the above-mentioned four test formulations (3 mL) was added to a vial (10 mL volume with 2 cm diameter) to sufficiently cover the base. The vial size was selected to reduce any potential surface tension influencing penetration into the test formulations. Carbon black nanopowder (0.35% *w*/*w*; Merck, Darmstadt, Germany) with particle size of <100 nm was selected to replicate the common size of 80 to 120 nm of a typical enveloped virus. The carbon black suspension (0.35% *w*/*w*) in water was then applied to the surface of each formulation, and the results were video-recorded over 12 min.

### 2.7. Cellullar Metabolism and Viability Assays

Cell viability was assessed using the AlamarBlue cell viability assay (Thermo Fisher, Scoresby, Australia) as previously described [26]. Cell viability was also assessed using the CyQUANT LDH cytotoxicity assay (Thermo Fisher Scientific, Waltham, MA, USA) as previously described [27]. Detailed procedures can be found in the Appendix A.

### 2.8. Measurement of Ciliary Beat Frequency of hNECs

Ciliary beating frequency (CBF), which is the rate of cilia beating per second, was auto-analyzed using the Sisson-Ammons video analysis (SAVA) system (Ammons Engineering, Clio, MI, USA) as previously described [26]. Detailed procedures are in the Appendix A.

### 2.9. Cytospin Preparation

The hNECs were dissociated into single cell suspensions and were fixed in 4% formaldehyde at room temperature for 10 min. Cells were washed thrice using 1× dPBS, centrifuged at 500 rpm for 5 min, and stored at −20 °C until staining.

### 2.10. Immunofluorescence Staining

Fixed cells on slides were used for performing immunofluorescence staining. The antibodies and staining procedures are described in the Appendix A.

### 2.11. Statistical Analysis

All results were analyzed using Prism 6 software (GraphPad, San Diego, CA, USA). The data were not normally distributed (Gaussian distribution) due to the variability among the different individual-derived hNECs that were analyzed by GraphPad. Consequently, non-parametric summary statistics and statistical tests were performed. The median values and interquartile ranges were used to describe the observed data. A non-parametric one-way ANOVA and Dunn’s multiple comparisons were tested at the 5% level of significance.

## 3. Results

### 3.1. HPMC Serves as an Effective Physical Barrier Against Diffusion of Methylene Blue and Nanoparticles

In less than 2 min after application, the diffusion of methylene blue through the membrane into the receiver fluid was observed in both the water only and pH 3.5 buffer only formulations (Figure 1A). The receiver fluid for the non-HPMC formulations turned completely blue after 12 min. In contrast, the receiver fluid remained colorless 12 min after application in both the 1% HPMC in water and 1% HPMC in pH 3.5 buffer formulations. The experiment demonstrated that 1% HPMC provides an effective physical barrier against methylene blue diffusion in both water and low pH conditions. Similarly, the solid particle barrier assessment illustrates a similar trend as the liquid barrier assessment (Figure 1B). In less than 2 min after application to the surface, the diffusion of carbon black nanopowder (<100 nm) to the bottom of the vial was observed in both the water only and pH 3.5 buffer formulations, and formed an opaque suspension in 5 min. Interestingly, the carbon black nanopowder was contained at the surfaces of 1% HPMC in water and 1% HPMC in pH 3.5 buffer formulations immediately after application (0 min). Once the agglomerated carbon particles reached a critical mass, they dropped to the bottom of the vial at 2 min. At 5 min, the 1% HPMC in water and 1% HPMC in pH 3.5 buffer formulations remained as clear solutions with agglomerated carbon particles found at the bottom of the vial. The experiment showed that 1% HPMC provides an effective physical barrier via particle agglomeration in both water and low pH conditions.

### 3.2. HPMC Treatment in Low pH Buffer Does Not Affect Cell Viability and Muco-Ciliary Function of hNECs

To examine the tolerability of hNECs to 0.5% HPMC and low acidity, 0.5% HPMC in 1× dPBS (pH 3.5 and pH 5) and 1× dPBS acidic buffer alone (pH 3.5 and pH 5) were added to hNECs for 1 day and for 1 week. The percentage reduction in resazurin of untreated hNECs (blank) and hNECs with 1× dPBS (pH 7) were also measured for comparison. There was no significant change in the percentage reduction in resazurin by total live cells; hence, no significant alteration in cellular metabolic activity was detected for hNECs treated with 0.5% HPMC at pH 3.5 and pH 5 and acidic buffer alone (1× dPBS at pH 3.5 and pH 5) (Appendix A). Additionally, cell viability was also assessed by the LDH assay using the apical supernatant of hNECs to detect LDH release by dead cells. Similarly, there was no significant change in the LDH released by dead cells for hNECs treated with 0.5% HPMC at pH 3.5 and pH 5 and acidic buffer alone (1× dPBS at pH 3.5 and pH 5) (Appendix A). CBF was measured to assess the effect of HPMC in acidic buffers on muco-ciliary function of hNECs. There was no significant reduction in CBF of hNECs following a one-day treatment of 0.5% HPMC at pH 3.5, suggesting that hNECs exhibited the ability to tolerate for up to one week (Appendix A).

### 3.3. Pre-Treatment with HPMC in Acidic Buffer Prior to Infection of hNECs Reduces IAV and RV Titers

Pre-treatment with 0.5% HPMC was administered to hNECs 4 h prior to infection (Appendix A). Preliminary tests were performed to assess the effects of different acidity and HPMC on H1N1 and RV titers during infection. H1N1 was chosen for the preliminary tests as it is a clinical isolate derived from a patient. The acidic buffer (pH 3.5) treatment alone pre- and post-infection reduced the viral titers of H1N1 and RV as compared to neutral buffer (pH 7) alone, suggesting that low acidity reduces viral production of infected hNECs. Treatment with HPMC in acidic buffer (pH 3.5) pre- and post-infection also slightly reduced the titers of H1N1 and RV as compared to HPMC in neutral buffer (pH 7), albeit without statistical significance (Appendix A). This suggests that HPMC in acidic and neutral buffer likely coats the hNECs to serve as a physical barrier, thereby hindering viral infection and production. The levels of total virus production for HPMC pre-treatment in pH 3.5 and pH 7 buffer before and after H1N1 and RV infection are presented in Appendix A. We have also examined the effects of different acidity and HPMC on the epithelial integrity of hNECs during H1N1 and RV infection by determining their trans-epithelial electrical resistance (TEER). There was no significant change in the TEER of the hNECs between treatment using the pH 3.5 buffer and pH 7 buffer with and without HPMC at 24 h following H1N1 and RV infection as compared to no treatment (blank control) (Appendix A). The TEER values are presented in Appendix A. We subsequently focused our study on investigating the effects of HPMC in acidic buffer against three common respiratory viruses, i.e., H3N2, H1N1, and RV.

Pre-treatment of hNECs using acidic buffer alone slightly reduced IAV titers, whereas pre-treatment of hNECs using HPMC in acidic buffer (pH 3.5) significantly reduced IAV titers (PFU reduction of 67.1% for H1N1; 74% for H3N2) as compared to no treatment (Figure 2A,B), suggesting an additive effect on the inhibition of IAV. Interestingly, pre-treatment of hNECs using acidic buffer alone or HPMC in acidic buffer (pH 3.5) significantly reduced RV infection (PFU reduction of 76.8% and 75.1%, respectively) as compared to no treatment (Figure 2C). The levels of total virus production for HPMC treatment in pH 3.5 buffer before and after H3N2, H1N1, and RV infection are presented in Appendix A. To assess the pH restored by physiological homeostasis of hNECs, the pH of PBS (at pH 3.5) and HPMC in acidic buffer (pH 3.5) was examined by the pH strip test before their administration onto hNECs and after 4 h of treatment on hNECs. We found that the pH of both treatment with acidic PBS and HPMC in acidic buffer was altered to pH of ~5 after 4 h of their administration onto hNECs (Appendix A).

### 3.4. Pre-Treatment of hNECs Using HPMC in Acidic Buffer Does Not Affect Ciliary Function of hNECs Infected with H1N1 and RV

The positive immunofluorescence (IF) staining of H1N1 NS1 and RV VP2 antigens validated the presence of active viral infections in hNECs following pre-treatment with acidic buffer alone and HPMC in acidic buffer (Figure 3A,B). As RV exclusively infects ciliated cells [28] while IAV infects both goblet cells and ciliated cells [29], the area of intensity for acetylated α-tubulin-positive staining (ciliated cells) in hNECs infected with H1N1 and RV was assessed in comparison to mock-infected hNECs for the respective treatments. In the presence or absence of treatment, there was no significant difference in the ratio of acetylated α-tubulin staining for H1N1 and RV infection of hNECs as compared to mock-infected hNECs. This suggests that the pre-treatment of acidic buffer alone and HPMC in acidic buffer did not significantly alter the cilia density as compared to the infected and non-treated control groups at 24 hpi (Figure 3C,D). There was also no significant change in CBF of the hNECs infected with H1N1 and RV subjected to treatment as compared to the CBF of infected hNECs without treatment (Figure 3E,F).

### 3.5. Post-Infection Treatment of hNECs Using HPMC in Acidic Buffer Reduces IAV and RV Titers

Treatment of hNECs using both HPMC in acidic buffer (pH 3.5) and acidic buffer alone significantly reduced IAV titers (PFU reductions of 77.6% for H1N1, 91% for H3N2; and 93% for H1N1; 91.4% for H3N2, respectively) as compared to without treatment (Figure 4A,B). Treatment of hNECs using acidic buffer alone at 4 hpi slightly reduced RV titer (26.9% PFU reduction) compared to no treatment, albeit without statistical significance. However, the RV titer was further diminished by treatment with HPMC in acidic buffer (pH 3.5) (69.9% PFU reduction) as compared to no treatment (Figure 4C), suggesting an additive effect of HPMC and acidic buffer on the inhibition of RV infection. The levels of total virus production for HPMC treatment in pH 3.5 buffer before and after H3N2, H1N1, and RV infection are presented in Appendix A.

### 3.6. Post-Infection Treatment of hNECs Using HPMC in Acidic Buffer Does Not Affect Ciliary Function of hNECs Infected with H1N1 and RV

The positive IF staining of H1N1 NS1 and RV VP2 antigens validated active viral infections of hNECs with post-infection treatment of acidic buffer alone and HPMC in acidic buffer (Figure 5A,B). There was no significant difference in the area of intensity for acetylated α-tubulin staining for H1N1 and RV infection of hNECs as compared to mock-infected hNECs, with or without treatment. For H1N1 and RV infections at 24 hpi, this suggests that the treatment with acidic buffer alone and HPMC in acidic buffer did not significantly alter the cilia density as compared to the infected and non-treated controls (Figure 5C,D). There was also no significant change to CBF of the hNECs infected with H1N1 and RV with treatment as compared to that without treatment (Figure 5E,F).

## 4. Discussion

HPMC is an odorless and tasteless powder that is a derivative of cellulose. It is widely used in clinical applications such as drug delivery to the eye, airway, colon, and skin. Our preliminary studies using 1% HPMC in pH 3.5 buffer (Vicks First Defence Nasal Spray) showed that both 1% HPMC in water and in pH 3.5 buffer effectively inhibited the diffusion of methylene blue dye (0.02% *w*/*w*) as compared to water alone and acidic buffer alone without HPMC. Additionally, 1% HPMC in pH 3.5 buffer was shown to have the least diffusion of carbon black nanopowder (0.35% *w*/*w*) as compared to other test formulations. Following the preliminary results, we proceeded to investigate the effects of HPMC in pH 3.5 buffer against respiratory viruses in the hNEC model to mimic real-life virus infections in the upper airway with nasal spray treatment (HPMC in acidic buffer) before and after infection.

The nasal epithelium is in constant contact with the external atmosphere and is the primary site of virus infection. Our previously established in vitro hNEC infection model allows repeatable experiments to be performed in a standardized and controlled manner. Hence, this model enables the investigation of the physical actions of HPMC which can serve as an intervention against respiratory infections. Here, we report the different patterns of altered viral loads of RV and IAV in hNECs with pre- and post-infection treatment using HPMC in acidic buffer. We first briefly assessed the effects of treatment with different acidity on virus infections of hNECs. The results showed that while acidic buffer alone reduced the viral titer during infection, HPMC in acidic or neutral buffer also decreased the viral load and dampened viral infection by serving as a physical barrier on hNECs during infection. HPMC together with a low-acidity buffer may potentially further diminish the viral load during infection. We have evaluated the effects of pre- and post-infection treatment using acidic buffer, HPMC in acidic buffer (pH 3.5), neutral buffer, and HPMC in neutral buffer (pH 7) on the epithelial integrity of hNECs during infection. The acidity of pre- and post-infection treatment induced no or minimal effects on the epithelial integrity and ciliary function of hNECs during infection as compared to without treatment. We thus focused our study on investigating the effects of HPMC in acidic buffer against three common respiratory viruses, i.e., H3N2, H1N1, and RV.

RV is acid-labile, unstable at acidic pH (≤5), and completely inactivated at pH < 3 [19]. Pre-treatment of hNECs using acidic buffer alone reduced RV progeny production likely by destabilizing the challenge virus before the entry of virus particles into the host cells. The pre-treatment using acidic buffer before RV infection transiently provides an acidic environment which may thus destabilize RV to prophylactically diminish RV infection of host cells. In addition, treatment with acidic buffer alone reduced the viral load of IAV. Studies have reported inactivation of influenza virus at pH 3.5, with strongest inactivation at pH 4.5, but no detectable virus inactivation at pH 5.5 [20,31]. It is noteworthy that the nasal pH can be rapidly restored through the acid-base homeostasis in the nasal cavity. The inconsistent virus reduction effect of acidic buffer is likely due to the restoration to higher physiological pH by the acid-base homeostasis of the hNECs. The nasal mucosal pH is approximately 5.5 to 6.5 [32], and its pH may be transiently lowered after intranasal administration of acidic buffer solution. Gern et al. [19] reported that intranasal administration of a pH 3.5 solution induced transient pH changes with the greatest pH reduction (2 to 2.5 pH units) observed within 1 min after administration, and the pH remained significantly lower than baseline pH for 5 to 10 min before reverting to baseline pH in healthy volunteers. Our study has validated that the pH of the supernatants of both acidic buffer and HPMC in acidic buffer increased from pH of ~3 to ~5 after 4 h of treatment on hNECs–this demonstrates that the acidity effect is temporary and highly regulated by the acid-base homeostasis of hNECs.

Compared to pre-treatment with acidic buffer alone, we observed an additive and stronger prophylactic effect which further diminished IAV progeny production when hNECs were pre-treated with HPMC in acidic buffer prior to exposure of hNECs to IAV. This suggests that HPMC may have exerted its primary mechanism of intervention by acting as a physical barrier against virus infection. Our diffusion assay using carbon black nanoparticles of <100 nm supports that HPMC likely serves as a physical barrier to impede the movement of IAV particles (~120 nm in size) and to restrict virus contact with the nasal epithelium for viral entry. Similarly, the minimal diffusion of methylene blue dye (molecular size of 1.6 nm × 0.7 nm) across HPMC also suggests that HPMC is effective as a physical barrier for smaller particles [33]. The primary protective action of HPMC is to serve as a physical barrier on the nasal epithelium which may hinder the release or lateral spread of virus progeny. Congruent with our findings, a previous study reported that HPMC treatment before and after SARS-CoV-2 infection effectively inhibited virus infection and virus release, respectively [11]. In comparison, our study revealed that pre-treatment using HPMC in acidic buffer resulted in an additional 40% reduction in IAV progeny release in hNECs as compared to acidic buffer alone.

At 4 h after the initial RV infection, the treatment of infected hNECs using HPMC in acidic buffer further restricted the subsequent production of RV progeny at 24 hpi, compared to treatment with acidic buffer alone. The HPMC treatment at 4 h after infection may have interfered with the early stage of RV replication. The coating of HPMC on the surface of host cells may hamper the release and intercellular spreading of RV on the apical surface of the nasal epithelium. Hence, HPMC may play a protective role by acting as a physical barrier on the nasal epithelium to mitigate the release or lateral spread of RV progeny after the initial virus infection.

Pre-treatment using acidic buffer alone slightly reduced IAV infection. On the other hand, post-infection treatment using either acidic buffer alone or HPMC in acidic buffer could significantly reduce IAV progeny production, which may be attributed to the low acidity environment. Hence, the treatment of hNECs using acidic buffer may exert an indirect mechanism in limiting IAV spread after initial virus infection of host cells. In the human airway, heavily glycosylated and sialylated mucins typically form a barrier against glycan-receptor-binding viruses, which are at risk of being trapped in the mucus layer and expelled by muco-ciliary clearance [34]. The mesh size of respiratory mucus ranges from 100 to 500 nm, and thus allows smaller virus particles such as non-enveloped RV to penetrate, while serving as a biological sieve to impede the movement of larger virus particles such as enveloped IAV [35]. To counteract this, IAV is capable of releasing its progeny virions by the action of IAV-associated glycan-receptor-destroying enzymes, i.e., neuraminidase [36]. It is reported that increased acidity on the airway mucosa leads to an increase in the viscosity of mucus [37,38]. Therefore, the treatment using acidic buffer may interfere with the spreading of IAV particles during the early stage of replication after the initial exposure to the virus.

Our study demonstrates the action of HPMC as a non-specific physical barrier in the physiologically relevant in vitro hNEC model; however, there are additional considerations for nasal spray application in vivo. Our study administered HPMC for 4 h on hNECs prior to virus infection, and at 4 h after initial infection to mimic application for early intervention when a person feels at risk of catching a cold. However, the natural process of muco-ciliary clearance in the nose will remove HPMC over time. Mucosal transport occurs at an average of 6 mm per min [39]. Our hNEC model represents the surface area of 33 mm^2^ of nasal epithelium. Therefore, the duration for the spread and clearance of products administered in the nose needs to be examined for the effectiveness of HPMC as a non-specific physical barrier in the upper airway. Additionally, the route of virus spread can potentially be via secretion from both the mouth and nose, as well as in droplets originating in the lungs. Hence, the complete coverage of HPMC by nasal administration to inhibit all sources of virus excretion would be challenging.

This study demonstrates the non-specific physical barrier effect of HPMC on the surface of hNECs as its primary action in protecting against respiratory virus infection. Interestingly, the acidic buffer may have a physical virus-destabilizing effect outside the hNECs, and its protective effect is additive when used in combination with HPMC. This formulation could thus potentially target a range of respiratory viruses including large, enveloped viruses and small and non-enveloped viruses, potentially serving as a prophylactic and therapeutic antiviral agent for the nasal epithelium.

## 5. Conclusions

Our study investigated the physical action of HPMC against common respiratory viruses using the physiologically relevant in vitro hNEC model which closely mimics the human upper respiratory airway. The physical barrier created by HPMC on the surface of the nasal epithelium serves as the primary protective action against both RV and IAV by impeding the diffusion of viruses to hNECs and by limiting the release or lateral spread of viruses. Administration of acidic buffer before or after virus infection may exert a physical virus-destabilizing effect outside the hNECs. The use of HPMC in combination with acidic buffer may serve as a prophylactic and therapeutic antiviral agent by targeting the spread of enveloped and non-enveloped respiratory viruses in hNECs.

## Figures and Tables

**Figure 1 viruses-17-00376-f001:**
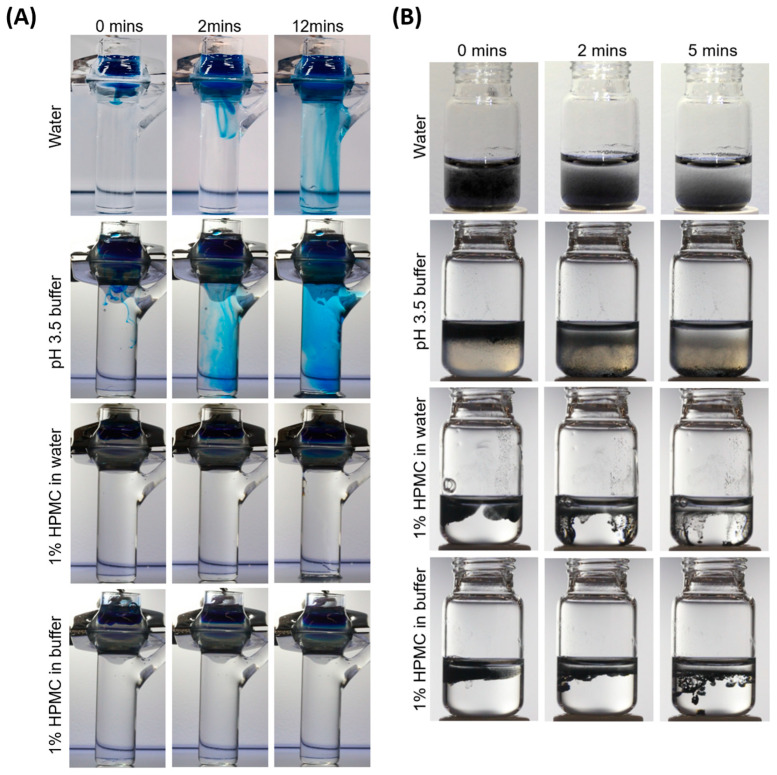
1% HPMC serves as an effective physical barrier against methylene blue and nanoparticle diffusion. (**A**) Diffusion of methylene blue dye through a nylon membrane, when applied directly to four formulations (water only, pH 3.5 buffer only, 1% HPMC in water, and 1% HPMC in pH 3.5 buffer) over a period of 12 min, with representative screenshots from the videos at 0 min (**left**), 2 min (**center**), and 12 min (**right**). (**B**) Diffusion of carbon black nanopowder (<100 nm) suspension over a period of 5 min, when applied directly to the surface of four test formulations (water only, pH 3.5 buffer only, 1% HPMC in water, and 1% HPMC in pH 3.5 buffer). Representative screenshots from the videos at 0 min (**left**), 2 min (**center**), and 5 min (**right**) are shown.

**Figure 2 viruses-17-00376-f002:**
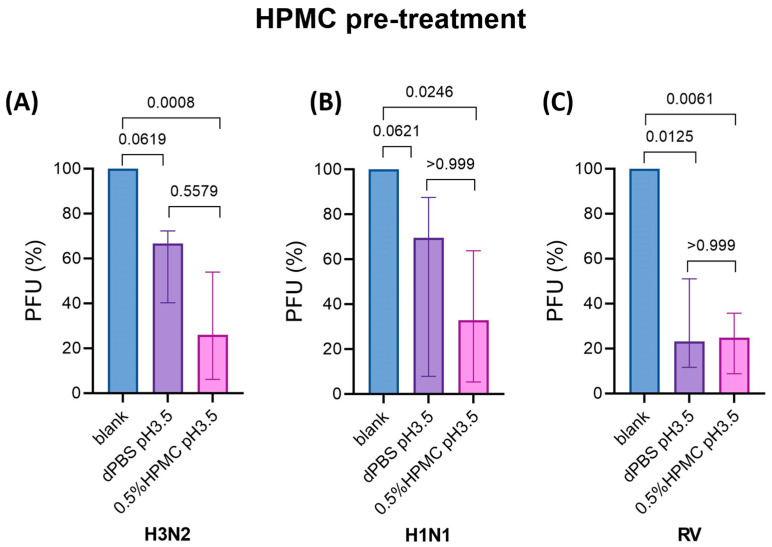
Pre-treatment of hNECs using HPMC in acidic buffer (pH 3.5) significantly reduced virus progeny production of IAV and RV at 24 hpi. (**A**) The pre-treatment of hNECs using acidic buffer alone (pH 3.5) slightly reduced the release of H3N2 progeny production, and the viral titer was further reduced by pre-treatment using 0.5% HPMC in acidic buffer (*n* = 6). (**B**) The pre-treatment of hNECs using acidic buffer alone slightly reduced the release of H1N1 progeny production, and the viral titer was further reduced by pre-treatment using 0.5% HPMC in acidic buffer (*n* = 6). (**C**) The pre-treatment of hNECs using acidic buffer alone and 0.5% HPMC in acidic buffer significantly reduced the release of RV progeny production as compared to the no treatment group (blank control) (*n* = 6).

**Figure 3 viruses-17-00376-f003:**
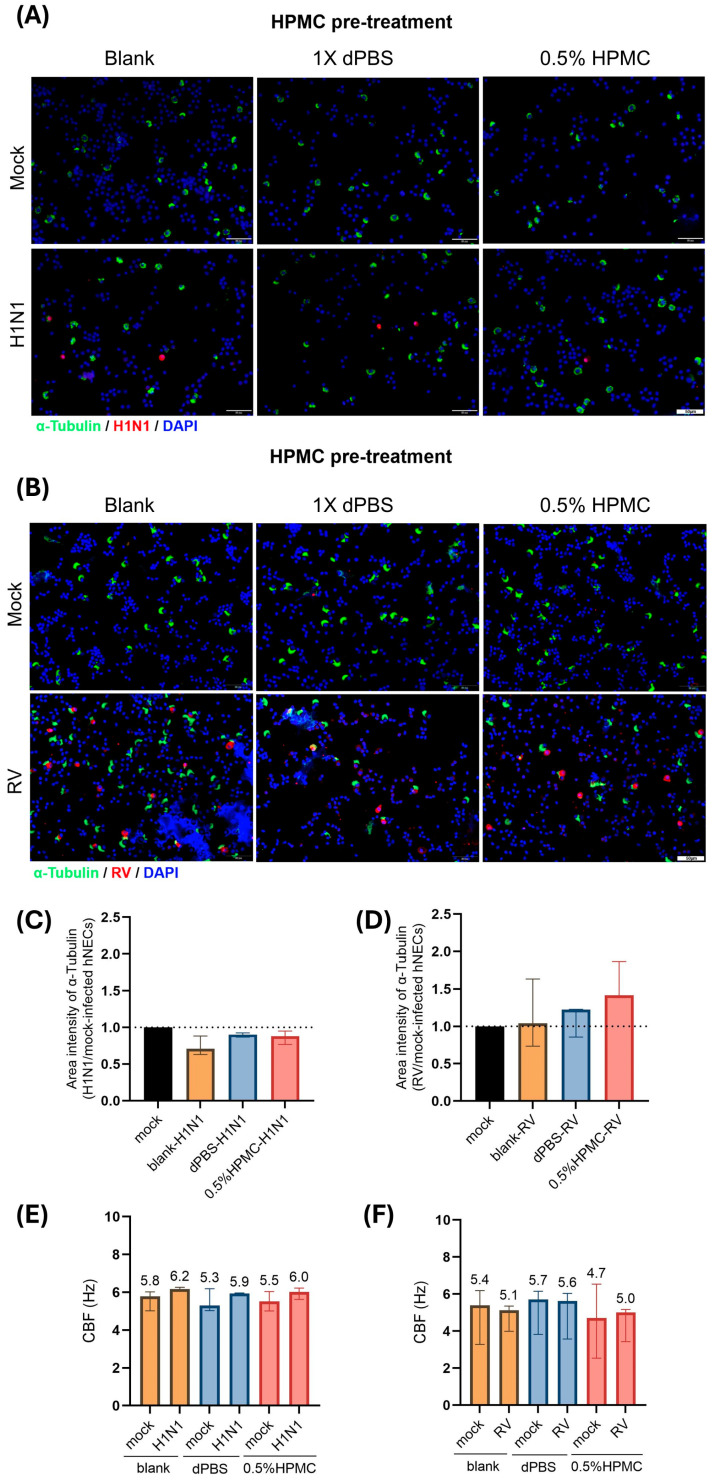
Pre-treatment with acidic buffer alone and with 0.5% HPMC in acidic buffer at 4 h prior to infection did not significantly alter ciliary function of hNECs during H1N1 and RV infection. (**A**,**B**) The positive IF staining of H1N1 NS1 and RV VP2 antigens validated the presence of active virus infection in hNECs following pre-treatment with acidic buffer alone and HPMC in acidic buffer. (**C**,**D**) There was no significant difference in the ratio of the area of acetylated α-tubulin-positive staining (ciliated cells) in hNECs infected with H1N1 and RV as compared to mock-infected hNECs for the respective treatments (*n* = 3). (**E**,**F**) No significant change in CBF of hNECs infected with H1N1 and RV was observed with and without pre-treatment (*n* = 3).

**Figure 4 viruses-17-00376-f004:**
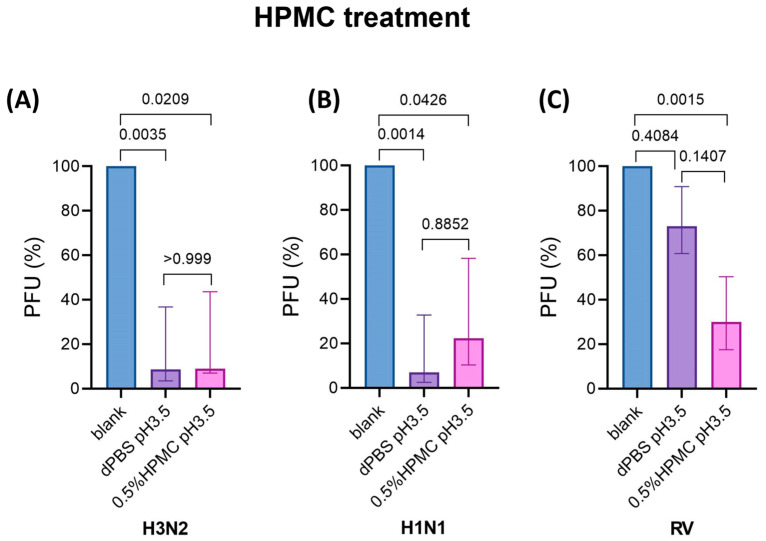
Post-infection treatment of hNECs using HPMC in acidic buffer significantly reduced virus progeny production of IAV and RV at 24 hpi. (**A**) The treatment of hNECs using acidic buffer alone (pH 3.5) and 0.5% HPMC in acidic buffer (pH 3.5) significantly reduced the release of H3N2 progeny production as compared to the non-treated control group (blank) (*n* = 6). (**B**) The treatment of hNECs using acidic buffer alone and 0.5% HPMC in acidic buffer significantly reduced the release of H1N1 progeny production as compared to the untreated control group (*n* = 6). (**C**) The treatment of hNECs using acidic buffer alone slightly reduced the release of RV progeny production, but the viral titer was further diminished by treatment using 0.5% HPMC in acidic buffer (*n* = 6).

**Figure 5 viruses-17-00376-f005:**
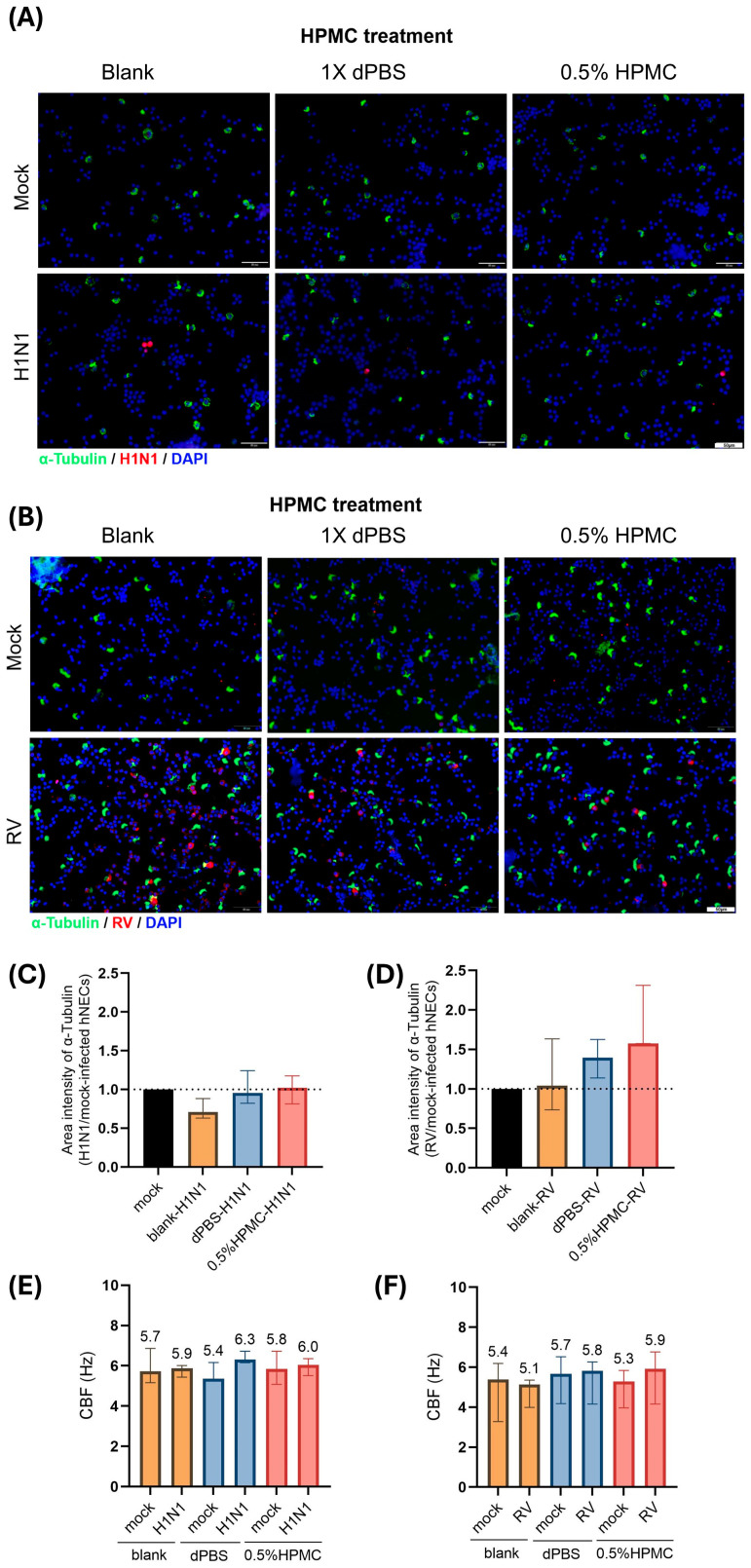
Post-infection treatment with acidic buffer alone and HPMC in acidic buffer at 4 hpi did not alter ciliary function of hNECs during H1N1 and RV infection. (**A**,**B**) The positive IF staining of H1N1 NS1 and RV VP2 antigens validated active virus infection of hNECs following treatment with acidic buffer alone and HPMC in acidic buffer. (**C**,**D**) There was no significant difference in the ratio of the area of acetylated α-tubulin-positive staining (ciliated cells) in hNECs infected with H1N1 and RV as compared to mock-infected hNECs for the respective treatments (*n* = 3). (**E**,**F**) No significant change in CBF of hNECs infected with H1N1 and RV infection was observed with and without treatment post-infection (*n* = 3).

## Data Availability

The raw data supporting the conclusions of this article will be made available by the authors on request.

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
