# Peer review of "Inhibitory Activity of Hydroxypropyl Methylcellulose on Rhinovirus and Influenza A Virus Infection of Human Nasal Epithelial Cells"

_viruses, 2025, doi:10.3390/v17030376_

Round 1
Reviewer 1 Report (Previous Reviewer 3)
Comments and Suggestions for Authors
The authors have taken into account the reviewer's suggestions made when the article was first submitted. Thank you!
Reviewer 2 Report (Previous Reviewer 2)
Comments and Suggestions for Authors
the revised manuscript has addressed all my comments adequately.
This manuscript is a resubmission of an earlier submission. The following is a list of the peer review reports and author responses from that submission.
Round 1
Reviewer 1 Report
Comments and Suggestions for Authors
The Focus of this study is to examine the effect of hydroxypropyl methylcellulose (HPMC) in acidic pH to act as barrier for respiratory viruses in nasal mucosa. Acidic pH has been examined for inactivation of virus previously and a new acidic HPMC formulation sold as Taffix has been shown to reduce respiratory viral infections when used twice a day in subjects at high risk (https://www.researchsquare.com/article/rs-100328/v1). Also this study did not report any adverse events in people using Taffix. Here authors demonstrate that 0.5% HPMC in acidic pH acts as a barrier for particulate material on a membrane and reduces viral load without affecting ciliary beat frequency using nasal epithelial cell cultures. The HPMC in pH 3.5 reduces the viral load similar to buffer alone. There are number of deficiencies in the manuscript.
1. The physiological pH in the nasal mucosa is around 6.5. This can be transiently reduced for a minute or so by spraying nasal sprays with acidic pH. However, the cell cultures may not be capable of neutralizing the acidic pH and therefore may overestimate the effect of HPMC in acidic pH.
2. Maintaining acidic pH on the apical surface of the cultures may quickly compromise barrier function by disrupting intercellular junctions which may actually increase the cellular invasion of virus. Here authors only show the shed viral load but not the intracellular virus, which in fact may be higher. The authors should show the effect of acidic HPMC on intercellular junctions and also cell associated viral load.
3. The acidic buffer alone (pH 3.5) reduces the influenza viral load similar to HPMC, but not for rhinovirus. While influenza is enveloped virus, rhinovirus is not. This should be discussed.
4. Normal Ciliary beat frequency in the nasal epithelial cultures should be between 11-15 Hz. Here authors report a mean value for CBF as 5.5, which is very low and it may not further decrease with acidic pH used in the study.
5. Antibody to acetylated alpha tubulin rather than alpha tubulin should be used to detect cilia.
6. Rhinovirus 16 used in this study bind to ICAM-1. In the scenario of compromised intercellular junctions, the basolaterally expressed ICAM-1 is readily accessible for virus to bind and enter the cells.
7. The manuscript needs extensive editing.
Minor comments
1. p3, ln 118: 1 g HPMC was dissolved in 100 ml 1X dPBS. (100 ml is missing).
2. 2X dilution is somewhat confusing. 0.5% HPMC is more clear
Comments on the Quality of English LanguageRequires editing
Reviewer 2 Report
Comments and Suggestions for Authors
Comments
The manuscript explores the use of topical HPMC administration with acidic buffer as a potential treatment for infection with RV and IAV using hNECs. The authors provide a comprehensive characterization of the impact of HPMC on hNEC viability and mucociliary function as well as its impact on virus production at 24 hours post infection.
Suggested experiments to further strengthen the arguments made in this study are as follows: 1) Include IFA of influenza A infection pre and post HPMC treatment. 2) Collect trans epithelial electrical resistance (TEER) readings pre- and post- infection and treatment to measure epithelial culture barrier integrity over the course of infection.
Major Comments
1. showing individual data points as scatter plots or violin plots is usually more appropriate than showing bar graphs with error bars.
2. Figure 2 and 4: Can the authors also include total PFUs as opposed to % reduction of PFUs as a metric of total virus production at 24 hours post infection?
Minor Comments
1. Figure 2: The authors should include an additional control with HPMC administered in a pH7 buffer to strengthen their argument that pre-treatment is most potent by both HPMC and low pH. Authors include a pH 7 blank in section 3.2 (Lines 206-207) for cell viability and mucociliary experiments.
2. Can the authors state how MOI was calculated? Is an MOI of 2.5 calculated upon the number of total cells in the hNEC well or the number of readily permissive cells at the time of infection.
3. Line 375 spelling error - "oof" should be of.
Reviewer 3 Report
Comments and Suggestions for Authors
The authors assessed the possibility of using HPMC for the prevention and treatment of respiratory viral infections (RV and IAV). Taking into account the reviewer's experimental suggestions, the manuscript deserves publication in Viruses MDPI.
Major:
1. To evaluate accurately the effect of HPMC on viral progenity, the reviewer suggests conducting experiments from paragraphs 3.3 and 3.5, for example, at pH 5 (HPMC pH 5.0).
This is especially necessary for comparing the results obtained at pH 5.0 with those of Figures 2C and 4C (for RV), in order to more clearly confirm the authors' assumption that:«The coating of HPMC on the surface of host cells may hamper the release and intercellular spreading of RV on the apical surface of the nasal epithelium. Hence, HPMC may play a protective role by serving as a physical barrier on the nasal epithelium to mitigate the release or lateral spread of RV progeny after the initial virus infection.»
Due to the small size of RV virions (approx. 30 nm in diameter), HPMC may not represent a barrier to free passage into and out of the cells.
2. The authors should either add a discussion of the experiment with 100 µm nylon filter scaffold membrane to section 4 or remove it from the manuscript.
3. According to figure S3 (HPMC treatment for 4 h) and experimental description (line 124): «For post-infection treatment, 100 µL of HPMC 2X was added directly onto hNECs at 4 hpi for 4 h. hNECs were then harvested at 24 hpi», HPMC treatment was carried out for 4 h.
please explain how the authors removed HPMC after 4 hours of treatment
Minor:
1. Please provide the definition of the abbreviation CBF when it first appears.
2. Line 52: «These commercial medical device nasal sprays» contains too many adjectives for nasal spray, please rewrite: These nasal sprays that are usually used with commercial …..
3. Line 430: «The hNECs were differentiated from human nasal epithelial stem/progenitor cells (hNESPCs) as described in previous papers [1, 2].» I looked through the two references provided and found nothing about differentiation. The authors should provide relevant references.
I also did not find the experiment details on Ciliary beating frequency (Measurement of ciliary beat frequency of hNECs, line 473) in [1].
However, in «2.8. Measurement of ciliary beat frequency of hNECs (line 151) » there is a reference to [26]:
«Ciliary beating frequency (CBF) was auto-analyzed using the Sisson-Ammons Video 152 Analysis system (SAVA, Omaha, NE, USA) as previously described. [26]»
Please, correct
Line 493: «Figure S1. 1 day and 1 week HPMC treatment in acidic buffer (pH 3.5 and pH 5) do not induced significant cell death on hNECs.» Resazurin Cell Viability Assay indicates the level of cell metabolism, and does not answer the question of whether they are alive or dead. Please, correct: Figure S1. 1 day and 1 week HPMC treatment in acidic buffer (pH 3.5 and pH 5) do not induced the decrease in cell metabolism and significant cell death on hNECs.
Line 375: correct «oof»